# Training a convolutional neural network to conserve mass in data assimilation

Yvonne Ruckstuhl[1], Tijana Janjić [1], and Stephan Rasp[2]

[1]Meteorological Institute Munich, Ludwig-Maximilians-Universität München, Germany
[2]ClimateAi, San Francisco, USA

**Correspondence:** Yvonne Ruckstuhl (yvonne.ruckstuhl@lmu.de)

**Abstract.** In previous work, it was shown that preservation of physical properties in the data assimilation framework can significantly reduce forecast errors. Proposed data assimilation methods, such as the quadratic programming ensemble (QPEns) that can impose such constraints on the calculation of the analysis, are computationally more expensive, severely limiting their application to high dimensional prediction systems as found in earth sciences. We therefore propose to use a convolutional neural network (CNN) trained on the difference between the analysis produced by a standard ensemble Kalman Filter (EnKF) and the QPEns to correct any violations of imposed constraints. In this paper, we focus on conservation of mass and show in an idealized setup that the hybrid of a CNN and the EnKF is capable of reducing analysis and background errors to the same level as the QPEns.

## 1 Introduction

The ensemble Kalman Filter (EnKF Evensen, 1994; Burgers et al., 1998; Evensen, 2009) and versions thereof are powerful data assimilation algorithms that can be applied to problems that need an estimate of a high dimensional model state, as in weather forecasting. An important condition for a successful application of the EnKF to a large system is the use of localisation. Any localisation method aims to diminish sampling errors caused by the computational limitation of the ensemble size. By doing so, mass conservation as guaranteed by a numerical model is violated during data assimilation (Janjić et al., 2014). It was shown in Janjić et al. (2014), Zeng and Janjić (2016), Zeng et al. (2017) and Ruckstuhl and Janjić (2018) that failing to conserve certain quantities like mass, energy and enstrophy can be highly detrimental to the estimation of the state. Janjić et al. (2014) proposed a new data assimilation algorithm, the Quadratic Programming Ensemble (QPEns), which replaces the analysis equations of the EnKF with an ensemble of minimisation problems subject to physical constraints. Zeng et al. (2017) showed in an idealised setup with a two week forecast generated by a two dimensional shallow water model that error growth is significantly reduced if the enstrophy is constrained. Similarly Ruckstuhl and Janjić (2018) illustrated the benefit of constraining the total mass and positivity of precipitation on a simple test case for convective scale data assimilation. The obstacle that remains for applying

the QPEns on large systems is the computational demand of solving the constrained minimisation problems that appear for each ensemble member at each assimilation cycle. For a detailed discussion on the computational costs of the QPEns we refer to Janjic et al. (accepted with minor revisions). In this work we propose to use an artificial neural network (NN) to correct the unconstrained solution, instead of solving the constrained minimisation problems.

NNs are powerful tools to approximate arbitrary nonlinear functions (Nielsen, 2015). A NN learns to recognize patterns based on examples, rather than being explicitly programmed. An important advantage is that no direct knowledge of the function is needed. Instead, a data set consisting of input-output pairs is used to train the NN to predict the output corresponding to a given input. Especially in the fields of image recognition and natural language processing, NNs are state-of-the-art and have become a standard tool (LeCun Yann et al., 2015). In numerical weather prediction NNs are not yet fully integrated, though interest is rising quickly (Reichstein et al., 2019). A recent review of the use of NNs in meteorology can be found in McGovern et al. (2019). Explored applications include (but are not limited to) post processing of raw model output based on observations (McGovern et al., 2017; Rasp and Lerch, 2018), representing subgrid processes in weather and climate models using high resolution model simulations (Krasnopolsky et al., 2013; Rasp et al., 2018; Brenowitz and Bretherton, 2019; Yuval and O'Gorman, 2020), combining a NN with a knowledge based model as a hybrid forecasting approach (Pathak et al., 2018b; Watson, 2019) and replacing the numerical weather prediction model all together (Dueben and Bauer, 2018; Pathak et al., 2018a; Weyn et al., 2020; Scher and Messori, 2019; Rasp et al., 2020; Rasp and Thuerey, 2020). A general challenge when applying NNs in numerical weather prediction is that the training data often consists of sparse and noisy data, which NNs are ill equipped to handle. Brajard et al. (2020a) and Bocquet et al. (2020) proposed to use data assimilation in the training process of the NN to deal with this issue. This approach has successfully been applied to reduce model errors (Brajard et al., 2020b; Farchi et al., 2020).

Fully replacing data assimilation by a NN has been attempted by Cintra and de Campos Velho (2014) in the context of a simplified atmospheric general circulation model. They trained on a cycling data set produced by the Local Ensemble Transform Kalman Filter (LETKF, Bishop et al., 2001; Hunt et al., 2007) and show that the trained NN performs nearly as good as the LETKF with significantly reduced computational effort. Other applications of NNs in context of data assimilation are for observational bias correction (Jin et al., 2019) and tuning of covariance localisation (Moosavi et al., 2019). In this paper we take an approach that combining the NN with a data assimilation algorithm will allow extracting the most information from sparse and noisy observations, as argued in for example Brajard et al. (2020a). We aim to produce better results than standard data assimilation algorithms at minimal additional computational costs, by training on data produced by the QPEns.

We generate our training data by performing twin experiments with the one dimensional modified shallow water model (Würsch and Craig, 2014) which was designed to mimic important properties of convection. These aspects include an acute regime switch when convection is triggered (conditional instability) and a significant time lag between the onset of convection and its observation. The model is briefly introduced in section 2.1, followed by the settings of the twin experiments in section 2.2. Section 2.3 provides a report on the generation of the training data. Since both our input and output are full model states, the obvious choice is to train a convolutional neural network (CNN), as the convolution with kernels naturally acts as a form of

localisation. The CNN architecture we use for this application is described in section 2.4. The results are presented in section 3, followed by the conclusion in section 4.

## 2   Experiment setup

### 2.1   Model

The modified shallow water model (Würsch and Craig, 2014) consists of the following equations for the velocity $u$, rain $r$ and water height level of the fluid $h$ respectively:

$$\frac{\partial u}{\partial t} + u\frac{\partial u}{\partial x} + \frac{\partial(\phi+\gamma^2 r)}{\partial x} = \beta_u + D_u\frac{\partial^2 u}{\partial x^2}, \tag{1}$$

with

$$\phi = \begin{cases} \phi_c & if \quad h > h_c \\ gh & \text{else}, \end{cases} \tag{2}$$

$$\frac{\partial r}{\partial t} + u\frac{\partial r}{\partial x} = D_r\frac{\partial^2 r}{\partial x^2} - \alpha r - \begin{cases} \delta\frac{\partial u}{\partial x}, & h > h_r \quad \text{and} \quad \frac{\partial u}{\partial x} < 0 \\ 0, & \text{else}, \end{cases} \tag{3}$$

$$\frac{\partial h}{\partial t} + \frac{\partial(uh)}{\partial x} = D_h\frac{\partial^2 h}{\partial x^2}. \tag{4}$$

Above, $h_c$ represents the level of free convection. When this threshold is reached the geopotential $\phi$ takes on a lower, constant value $\phi_c$. The parameters $D_u$, $D_r$, $D_h$ are the diffusion constants corresponding to $u, r, h$, respectively. Coefficient $\gamma := \sqrt{gh_0}$ is the gravity wave speed for the absolute fluid layer $h_0$ ($h_0 < h_c$). The small Gaussian shaped forcing $\beta_u$ is added at random locations to the velocity $u$ at every model time step. This is done in order to trigger perturbations that lead to convection. Parameters $\delta$ and $\alpha$ are the production and removal rate of rain respectively. When $h$ reaches the rain threshold $h_r$ ($h_r > h_c$), rain is 'produced', leading to a decrease of the water level and of buoyancy. The model conserves mass, so the spatial integral over $h$ is constant in time.

The one dimensional model domain, representing 125 km is discretised with $n = 250$ points, yielding the state vector $\mathbf{x} = [\mathbf{u}^T \mathbf{h}^T \mathbf{r}^T]^T \in \mathbb{R}^{750}$. The time step is chosen to be 5 seconds. The forcing $\beta_u$ has a Gaussian shape with half width of 4 grid points and an amplitude of 0.002 m/s. This model was used for testing data assimilation methods in convective scale applications in Haslehner et al. (2016); Ruckstuhl and Janjić (2018).

## 2.2 Twin experiments

The nature run which mimics the true state of the atmosphere is a model simulation starting from an arbitrary initial state. The ensemble is chosen to be of small size with $N_{ens} = 10$, and, like the nature run, each member starts from an arbitrary initial state. Observations are assimilated every $dT$ model time steps and are obtained by adding a Gaussian error to the wind $u$ and height $h$ field of the nature run at the corresponding time with a standard deviation of $\sigma_u = 0.001$ m/s and $\sigma_h = 0.01$ m, and a lognormal error is added to the rain $r$ field with parameters of the underlying normal distribution $\mu = -8$ and $\sigma = 1.5$. For all variables the observation error is roughly $10\%$ of the maximum deviation from the variable mean. To mimic radar data, observations for all variables are available only on grid points where rain above a threshold of $0.005$ dBZ is measured. A random selection, amounting to $10\%$ of the remaining grid points, of additional wind observations are assimilated, which represents additional available data (for example obtained from aircraft).

To deal with undersampling, covariance localisation using 5-th piecewise rational function (Gaspari and Cohn, 1999) is applied with a localisation radius of four grid points. This corresponds to the localisation radius for which the EnKF yields minimum analysis RMSE values of the rain variable for an ensemble size of ten. An interior point method is used to solve the quadratic minimisation problems of the QPEns. The constraints that are applied are mass conservation, i.e. $\mathbf{e}^T(\mathbf{h}^a - \mathbf{h}^b) = \mathbf{e}^T\delta\mathbf{h} = 0$, and positivity of precipitation, i.e. $\mathbf{r}^a = \delta\mathbf{r} + \mathbf{r}^b \geq 0$. Here, the superscript $b$ denotes the background and $a$ the analysis, and $\mathbf{e}$ is a vector of size $n$ containing only values of one. For the EnKF negative values for rain are set to zero if they occur.

When the assimilation window $dT$ is large enough, the accumulation of mass leads to divergence for the EnKF, that is, the analysis error is larger than the climatological standard deviation of the model state. The QPEns converges for all $dT$, due to its ability to conserve mass. We therefore distinguish two cases, one where the EnKF converges ($dT = 60$, equivalent to 5 minutes real time), and one where the EnKF diverges ($dT = 120$, equivalent to 10 minutes real time). We refer to Ruckstuhl and Janjić (2018) for a comparison of the performance of the EnKF and the QPEns as a function of ensemble size for different localisation radii, assimilation windows and observation coverage.

## 2.3 Training data

We aim to produce initial conditions of the same quality as the ones produced by the QPEns by upgrading the initial conditions produced by the EnKF using a CNN. To that end, we generate QPEns cycling data $\{(\mathbf{Q}_t^b, \mathbf{Q}_t^a) : t = 1, 2, ..., T\}$, where $\mathbf{Q}$ stands for QPEns, the superscript $b$ denotes the background and $a$ the analysis. In parallel we create the data set $\{\mathbf{X}_t^a : t = 1, 2, ..., T\}$, where $\mathbf{X}_t^a$ is the unconstrained solution calculated from $\mathbf{Q}_t^b$. See Figure 1 for a schematic of the generation process of the data sets. Note that by using the background generated from the QPEns ($\mathbf{Q}_t^b$) in the calculation of both $\mathbf{X}_t^a$ and $\mathbf{Q}_t^a$, we train the CNN only to focus on differences in the minimisation process and not on the possible differences in the background error covariances that could have accumulated during cycling. In section 3 we validate this approach by applying the CNN to the EnKF analysis for 180 subsequent data assimilation cycles. Both data sets contain the entire ensemble of $N_{ens} = 10$ members, such that $(*)_t^{(*)} \in \mathbb{R}^{N_{ens} \times n \times 3}$, where the last dimension represents the 3 variables $(u, h, r)$ and $n$ is the number of grid points.

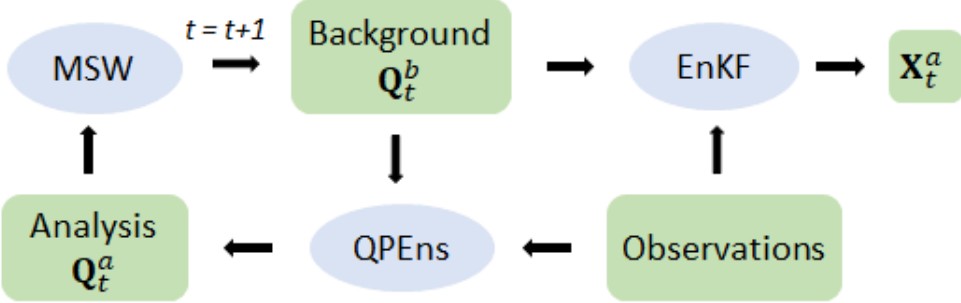

**Figure 1.** Schematic of the generation of the data sets $\mathbf{Q}_t^b$, $\mathbf{Q}_t^a$ and $\mathbf{X}_t^a$, where MSW stands for modified shallow water model.

The output of our training set $\mathbf{Y}^{tr} \in \mathbb{R}^{N_{ens}T \times n \times 3}$ is simply a reshaped and normalized version of the data set $\{\mathbf{Q}_t^a : t = 1, 2, ..., T\}$. For the input of our training set $\mathbf{X}^{tr}$ we choose to use an index vector indicating the position of the radar observations $\{\mathbf{I}_t : t = 1, 2, ..., T\}$ in addition to the unconstrained solutions $\{\mathbf{X}_t^a : t = 1, 2, ..., T\}$, yielding $\mathbf{X}^{tr} \in \mathbb{R}^{N_{ens}T \times n \times 4}$, where the index vector $\mathbf{I}_t$ is copied $N_{ens}$ times to obtain $\mathbf{I}_t^* \in \mathbb{R}^{N_{ens} \times n \times 1}$. We include this information because we know from Ruckstuhl and Janjić (2018) that the strength of the QPEns lies in suppressing spurious convection. Since the radar observations cover only rainy regions, the data set $\mathbf{I}_t$ can help the CNN to distinguish between dry and rainy regions and possibly develop a different regime for each situation. We verified that the CNN yields significantly different output when setting all values of $\mathbf{I}_t$ to zero, indicating that the CNN indeed uses this information. For $u$ and $h$ the input and output data set is normalized by subtracting the climatological mean before dividing by the climatological standard deviation. For $r$, we do not subtract the climatological mean to maintain positivity.

A validation data set $\mathbf{X}^{valid}$ and $\mathbf{Y}^{valid}$ exactly as the training data set but with a different random seed number is created to monitor the training process. For both the training and validation data set we set $T = 4800$, which amounts to a total of $N_{ens}T = 48000$ training and validation samples respectively.

### 2.4 Convolutional neural network architecture

We choose to use a CNN with 4 convolutional hidden layers, consisting of 32 filters each with kernels of size 3 and the "selu" activation function

$$g(x) = \lambda_1 \begin{cases} x, & \text{for } x \geq 0 \\ \lambda_2 \left( e^x - 1 \right), & \text{for } x < 0 \end{cases} \tag{5}$$

where $\lambda_1 = 1.05070098$ and $\lambda_2 = 1.67326324$. These values are chosen such that the mean and variance of the inputs are preserved between two consecutive layers (Klambauer et al., 2017). The output layer is a convolutional layer as well, where the number of filters is determined by the desired shape of the output of the CNN, which is a model state $(u, h, r) \in \mathbb{R}^{n \times 3}$. The output layer has therefore 3 filters and the kernel size is again 3. Note that the "localisation radius", that is, the maximum

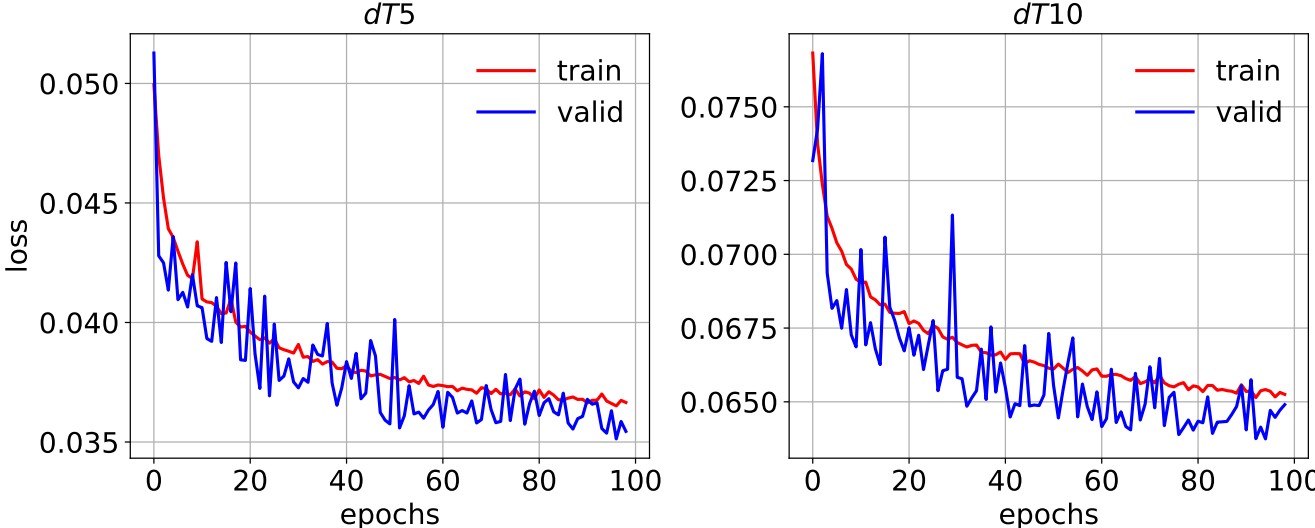

**Figure 2.** Value of the loss function $J$ averaged over samples for the training (red) and validation (blue) data set as function of epochs for $dT5$ (left) and $dT10$ (right).

influence radius of a variable as assumed by the CNN is $(3-1)/2*5 = 5$, where 5 is the number of layers and 3 the kernel size. We use a linear activation function for $u$ and $h$ and the "relu" activation function for $r$ to ensure non-negativity of rain. We set the batch size to 96 and run 100 epochs. Unless stated otherwise, the loss function is defined as the root mean squared error (RMSE) over the grid points, averaged over the variables:

$$J\left(\mathbf{y}_j^p(\mathbf{w})\right) = \frac{1}{3}\sum_{v=1}^{3}\sqrt{\frac{1}{n}\sum_{i=1}^{n}\left(y_{j,i,v}^p - y_{j,i,v}\right)^2}, \ \ j=1,\dots,N_{ens}T \tag{6}$$

where $y_{j,i,v}^p$ and $y_{j,i,v}$ are the prediction and output for the $v^{th}$ variable of the $j^{th}$ sample at the $i^{th}$ grid point respectively. The Adam algorithm is used to minimize $\frac{1}{N_{ens}T}\sum_{j=1}^{N_{ens}T} J\left(\mathbf{y}_j^p(\mathbf{w})\right)$ over the weights $\mathbf{w}$ of the CNN. The training is done with python library Keras (Chollet, 2017).

## 3  Results

We assign the name $dT5$ to the experiment corresponding to a cycling period of 5 minutes, and $dT10$ to the experiment corresponding to a cycling period of 10 minutes. Figure 2 shows the evolution of the loss function averaged over the samples for the training and validation data set for $dT5$ and $dT10$ respectively. Table 1 summarizes what the CNN has learned for each variable separately in the two cases. As the training data is normalized, we can conclude from the RMSE of the input data with respect to the output data (first row in Table 1 panels) that the mass constraint on $h$ and the positivity constraints on $r$ impacts the solution of the minimization problem for all variables with the same order of magnitude. Given our choice of loss function

| | Validation | loss | u | h | r | mass h | mass r | bias h |
|---|---|---|---|---|---|---|---|---|
| | Input | 4.9e-2 | 4.2e-2 | 4.6e-2 | 5.9e-2 | 1.8e-2 | 4.0e-3 | 1.4e-2 |
| $dT5$ | Prediction | 3.6e-2 | 4.1e-2 | 3.9e-2 | 2.7e-2 | 1.4e-2 | 2.9e-3 | 0.0 |
| | Improvement (%) | 27 | 2.8 | 15 | 55 | 22 | 28 | 100 |
| | Input | 9.6e-2 | 7.9e-2 | 9.0e-2 | 1.2e-1 | 3.6e-2 | 8.0e-3 | 3.3e-2 |
| $dT10$ | Prediction | 6.5e-2 | 7.3e-2 | 6.9e-2 | 5.4e-2 | 2.9e-2 | 7.1e-3 | 2.3e-2 |
| | Improvement (%) | 32 | 7.8 | 24 | 55 | 20 | 11 | 30 |

**Table 1.** The loss function, the mean RMSE of the variables $u$,$h$,$r$, the absolute mass error divided by the number of grid points $n$ for $h$ and $r$, and the bias of $h$ (columns) calculated for the input $\mathbf{X}^{valid}$ (top row) and the CNN prediction (middle row) with respect to the output $\mathbf{Y}^{valid}$ for the validation data sets. The last row shows the improvement of the prediction towards the output compared to the input in percentage. The top table corresponds to $dT5$, the bottom table to $dT10$.

it is not surprising that the relative reduction of the gap between the input and output by the CNN is proportional to the size of the gap. By aiming to minimize the mean RMSE of all variables, the CNN reduced the violation of the mass constraint by about 20% for both experiments. However, for $dT5$ the reduction in the bias of the height field is $100\%$, while for $dT10$ it is a mere $30\%$.

    Next, we are interested in how the CNNs perform when applied within the data assimilation cycling. In Figure 3, we
compare the performance of the EnKF, QPEns and the hybrid of CNN and EnKF, where CNN is applied as correction to the initial conditions computed by the EnKF. To avoid having to train a CNN for the spin-up phase where the increments are larger, we start the data assimilation for the EnKF and the CNN from the initial conditions produced by the QPEns at the $20^{th}$ cycle. The RMSEs shown in Figure 3 are calculated through time against the nature run for both the background and the analysis.

    With respect to RMSEs, for $dT5$ the CNN performs as well as the QPEns, despite having learned during training only
27% of the difference between the EnKF and QPEns analysis in terms of the loss function. For $dT10$ the CNN does perform significantly better than the EnKF, but clearly remains inferior to the QPEns. Given that in terms of the RMSE over the grid points, the CNN for $dT10$ is slightly better than the one for $dT5$, we hypothesize that the key to good performance of the CNN applied within the data assimilation cycling lies with preventing the accumulation of mass in $h$. When mass accumulates in clear regions, that is regions where for the nature run holds $h < h_c$, it has a snowball effect not only on $h$ itself but also on $r$,
see Figure 4. After all, clouds, and later rain, are produced whenever $h > h_c$. For $dT5$ the CNN does not score much better than for $dT10$ in terms of absolute mass error. However it was able to effectively remove all bias in $h$ (with a residual of $\mathcal{O}(10^{-5})$), in contrast to the CNN for $dT10$.

    To support this claim we trained an additional CNN with the training set corresponding to $dT = 10$, with a penalty term for the mass of $h$ in the loss function:

$$170 \quad \hat{J}_\eta \left( \mathbf{y}_j^p(\mathbf{w}) \right) = J \left( \mathbf{y}_j^p(\mathbf{w}) \right) + \frac{\eta}{n} \left| \sum_{i=1}^n y_{j,i,2}^p - \sum_{i=1}^n y_{j,i,2} \right| \tag{7}$$

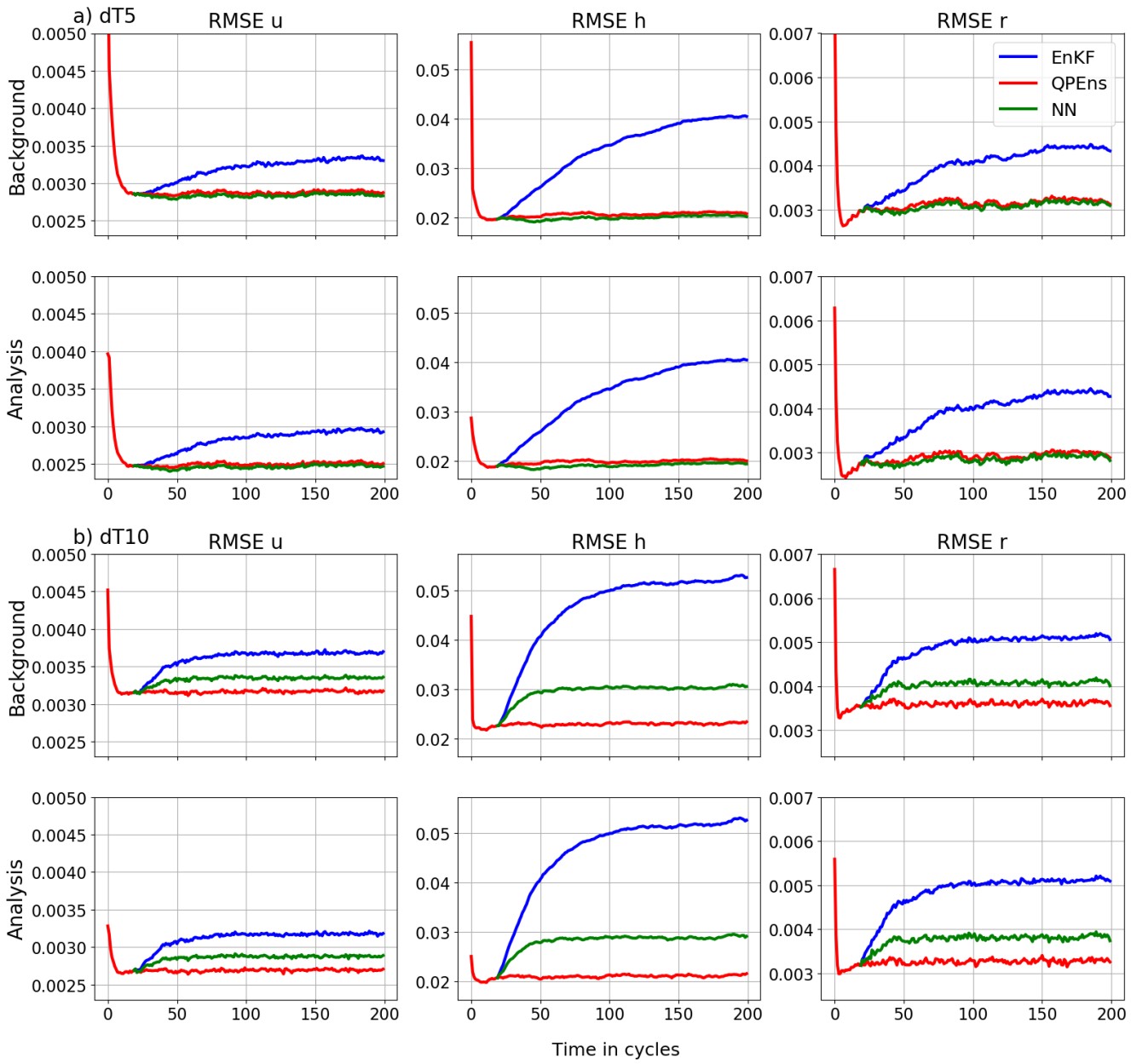

**Figure 3.** RMSE of the ensemble averaged over 500 experiments of the variables (columns) for the background (top rows) and analysis (bottom rows) as function of assimilation cycles for the EnKF (blue), the QPEns (red) and the CNN (green). The panels in a) corresponds to $dT5$ and in b) to $dT10$.

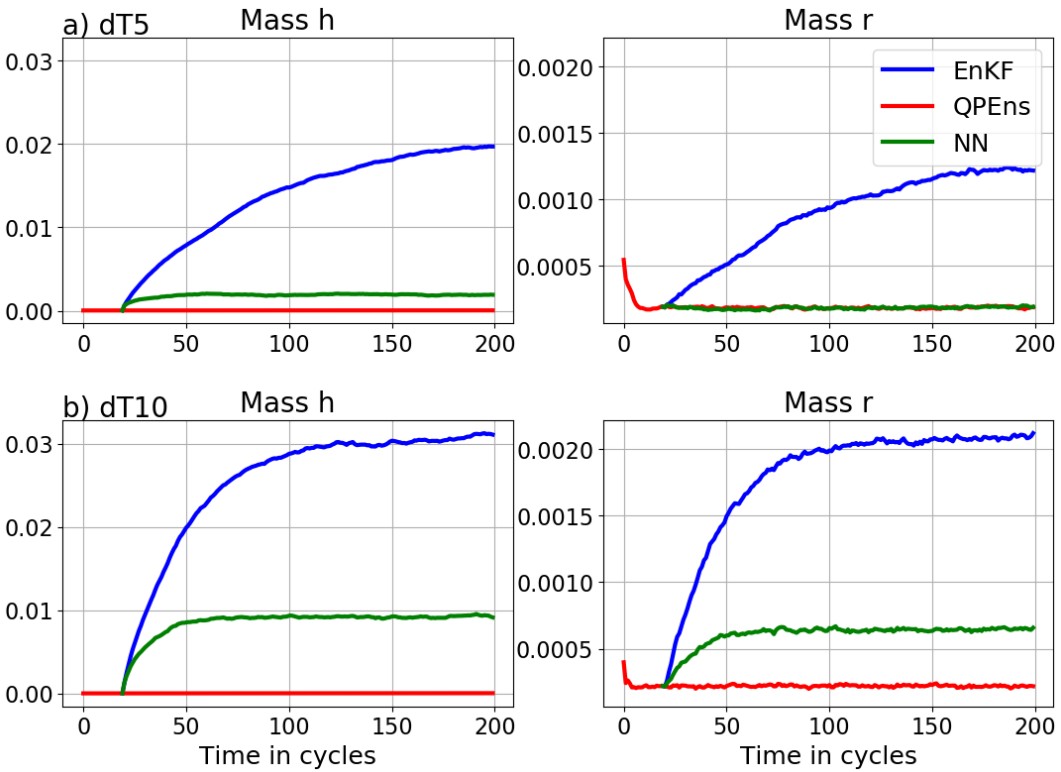

**Figure 4.** Absolute mass error averaged over 500 experiments of **h** (left) and **r** (right) for the analysis ensemble mean for the EnKF (blue), QPEns (red) and CNN (green). The plots in a) correspond to $dT5$ and in b) to $dT10$.

where the parameter $\eta$ is tunable. The larger $\eta$, the better the mass of $h$ is conserved at the expense of the RMSE, see Figure 5. We found a good trade-off for $\eta = 2$. We refer to this experiment as $dT10_{\eta=2}$. The training process is illustrated in Figure 5. The mass conservation term comprises about 40% of the total loss function $\hat{J}$. Both terms of the loss function are decreasing at approximately the same rate throughout the entire training process. Comparing Table 1 with Table 2 we conclude that by

adding the penalty term for the mass violation in the loss function, 7% of improvement was lost in terms of loss function $J$, but 29% was gained in the conservation of mass. Table 3 suggests that the CNN is especially active in clear regions or at the edge of clear regions. Indeed, by far the most significant correlations are with $h$, $r$ and $\frac{dh}{dx}$, where the negative sign indicates that the CNN corrects more in clear regions than in cloudy regions.

     Figures 6, 7 and 8 show the data assimilation results for $dT10_{\eta=2}$. It is striking that the CNN performs slightly better than the

QPEns. Since the CNN only has an influence radius of 5 grid points and the localisation cut-off radius of the data assimilation is 8 grid points, it is possible that the better results of the CNN stem from this shorter influence radius. However, a CNN trained on the same data but with kernel sizes of 5 instead of 3 (leading to an influence radius of 10 grid points) yields similar results as in Figures 6 and 7 (not shown). When comparing the input $\mathbf{X}$, output $\mathbf{Y}$ and the CNN prediction $\mathbf{Y}^p$ to the nature run, we found that for the clear regions $\mathbf{Y}^p$ is slightly closer to the nature run in terms of RMSE than the QPEns and significantly closer than

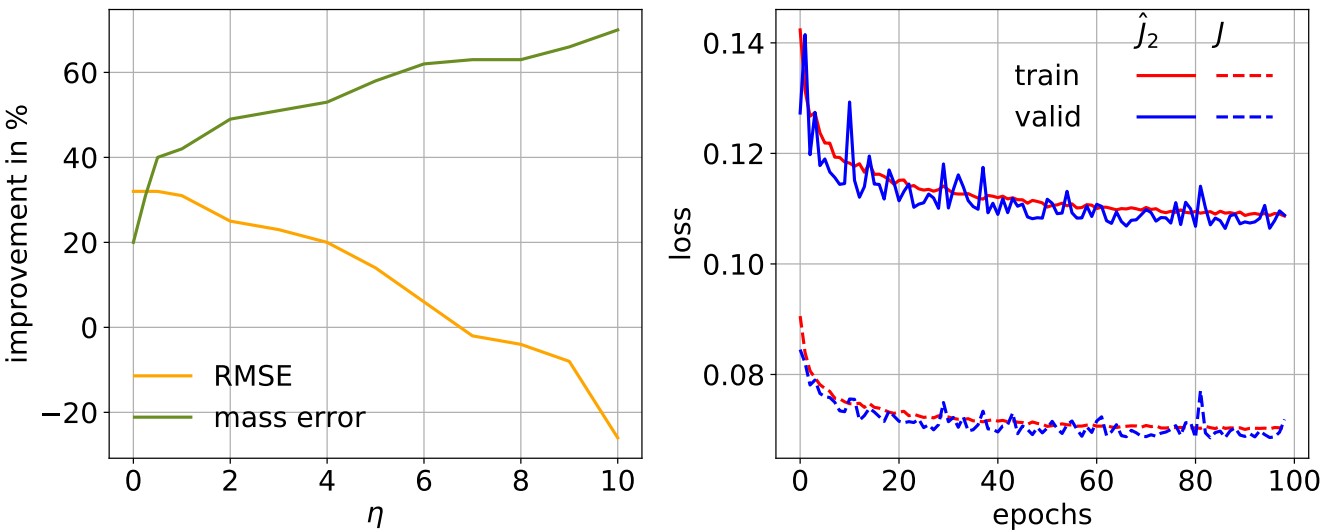

**Figure 5.** Left: relative improvement in % of RMSE (orange) and mass error (green) towards the output with respect to the input as a function $\eta$. Right: value of the loss function $\hat{J}$ (solid) and $J$ (dashed) averaged over samples for the training (red) and validation (blue) data set as function of epochs for $dT10_{\eta=2}$.

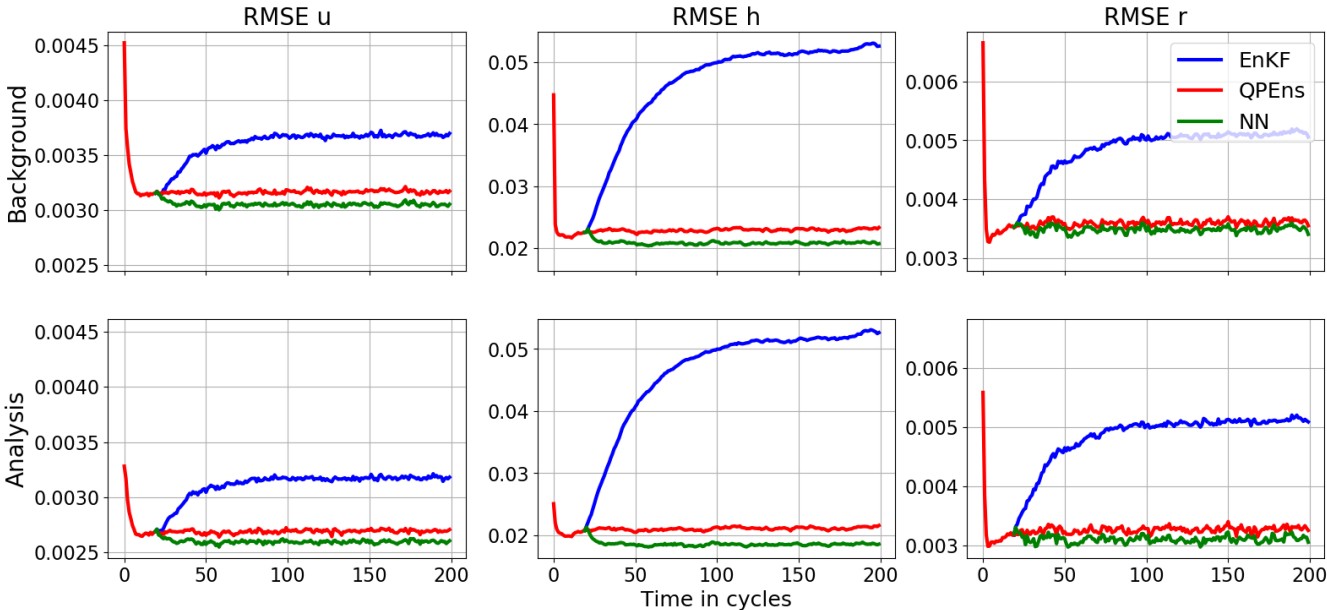

**Figure 6.** Same as Figure 3, but for $dT10_{\eta=2}$.

| validation | loss | u | h | r | mass h | mass r | bias h |
|---|---|---|---|---|---|---|---|
| Input | 9.6e-2 | 7.9e-2 | 8.9e-2 | 1.2e-1 | 3.6e-2 | 7.8e-3 | 3.3e-2 |
| Prediction | 7.2e-2 | 7.9e-2 | 8.2e-2 | 5.5e-2 | 1.8e-2 | 7.9e-3 | 8.9e-4 |
| Improvement (%) | 25 | 0.7 | 8.1 | 53 | 49 | -1 | 103 |

**Table 2.** Same as Table 1, but for $dT10_{\eta=2}$.

| | | $\mathbf{Y} - \mathbf{X}$ | | | $dT10: \mathbf{Y}^p - \mathbf{X}$ | | | $dT10_{\eta=2}: \mathbf{Y}^p - \mathbf{X}$ | | |
|---|---|---|---|---|---|---|---|---|---|---|
| | | **u** | **h** | **r** | **u** | **h** | **r** | **u** | **h** | **r** |
| **X** | **u** | -0.1 | 0.0 | 0.0 | -0.2 | 0.0 | 0.0 | -0.1 | 0.1 | 0.0 |
| | **h** | 0.0 | -0.1 | -0.1 | 0.1 | -0.2 | -0.2 | 0.2 | -0.5 | -0.2 |
| | **r** | 0.0 | -0.1 | -0.3 | 0.1 | -0.2 | -0.4 | 0.1 | -0.4 | -0.4 |
| $\frac{d\mathbf{X}}{dx}$ | **u** | -0.1 | 0.0 | 0.0 | -0.2 | 0.1 | 0.0 | -0.2 | 0.1 | 0.0 |
| | **h** | -0.2 | -0.1 | -0.2 | -0.4 | -0.2 | -0.2 | -0.4 | -0.2 | -0.2 |
| | **r** | -0.1 | -0.1 | -0.3 | -0.2 | -0.2 | -0.3 | -0.2 | -0.1 | -0.3 |

**Table 3.** Correlation coefficient for increments of the output (left) and the prediction for $dT10$ (middle) and $dT10_{\eta=2}$ (right) with the input (top) and the gradient of the input (bottom).

the EnKF (not shown). We speculate that this is because the QPEns generally lacks mass in regions where there are no clouds in both the nature run and the QPEns estimate. The EnKF on the other hand, overestimates the mass in these regions. This is clearly visible in the snapshot of Figure 8. As a result, the true value of $h$ lies between the QPEns and EnKF estimates. In these regions it is therefore favourable that the CNN can not completely close the gap between the input and output data, as it leads to a better fit to the nature run. We also performed an experiment where $h$ is updated by the CNN and the other variables remain equal to the EnKF solution, and similar results were obtained as in Figure 6 and 7. When only the clear regions of $h$ are updated by the CNN, the positive influence of the CNN is slightly reduced, but it still matches the performance of the QPEns. We therefore conclude that the success of this approach lies in the ability of the CNN to correct for errors of $h$, especially in clear regions.

## 4 Conclusion

Geoscience phenomena have several aspects that are different from standard data science applications, for example governing physical laws, noisy observations that are non-uniform in space and time from many different sources, and rare interesting events. This makes the use of NNs particularly challenging for convective scale applications, although attempts have been made for predicting rain, hail or tornadoes (McGovern et al., 2019). The approach taken in this study, combines noisy and

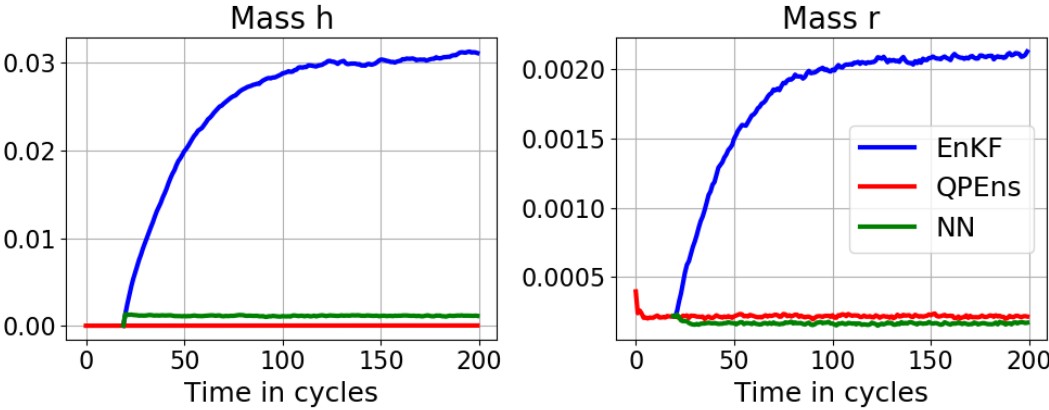

**Figure 7.** Same as Figure 4, but for $dT10_{\eta=2}$.

sparse observations with a dynamical model using a data assimilation algorithm, and in addition uses a CNN to improve on conservation of physical laws. In previous work it was shown in idealised setups that conserving physical quantities like mass in the data assimilation framework using the QPEns can significantly improve the estimate of the nature run. Here we show that it is possible to obtain similar positive results by training a CNN to conserve mass in a weak sense. By training on the unconstrained (EnKF)/constrained (QPEns) input/output pair, the CNN is able to reduce the mass violation significantly. Moreover, we found that adding a penalty term for mass violation in the loss function is necessary in one of the two test cases to produce data assimilation results that are as good as those corresponding to the QPEns.

These encouraging results prompt the question of the feasibility of this approach applied to fully complex numerical weather prediction systems. The challenge here lies in the generation of the training data. First, the effectiveness of conserving different quantities has to be verified in a non-idealised numerical weather prediction framework, where the quantities to be conserved may not be known and may not be exactly conserved within the numerical weather prediction model (Dubinkina, 2018). A second consideration is the computational costs. Advances are made in this regard (Janjic et al., accepted with minor revisions), but effort and collaboration with optimisation experts is still required to allow the generation of a reasonably large training data set.

*Code availability.* https://github.com/wavestoweather/MSW_DA_ML https://zenodo.org/badge/latestdoi/321384546

*Author contributions.* All of the authors contributed to research behind this paper, as well as to writing of the manuscript.

*Competing interests.* The authors declare that they have no conflict of interest.

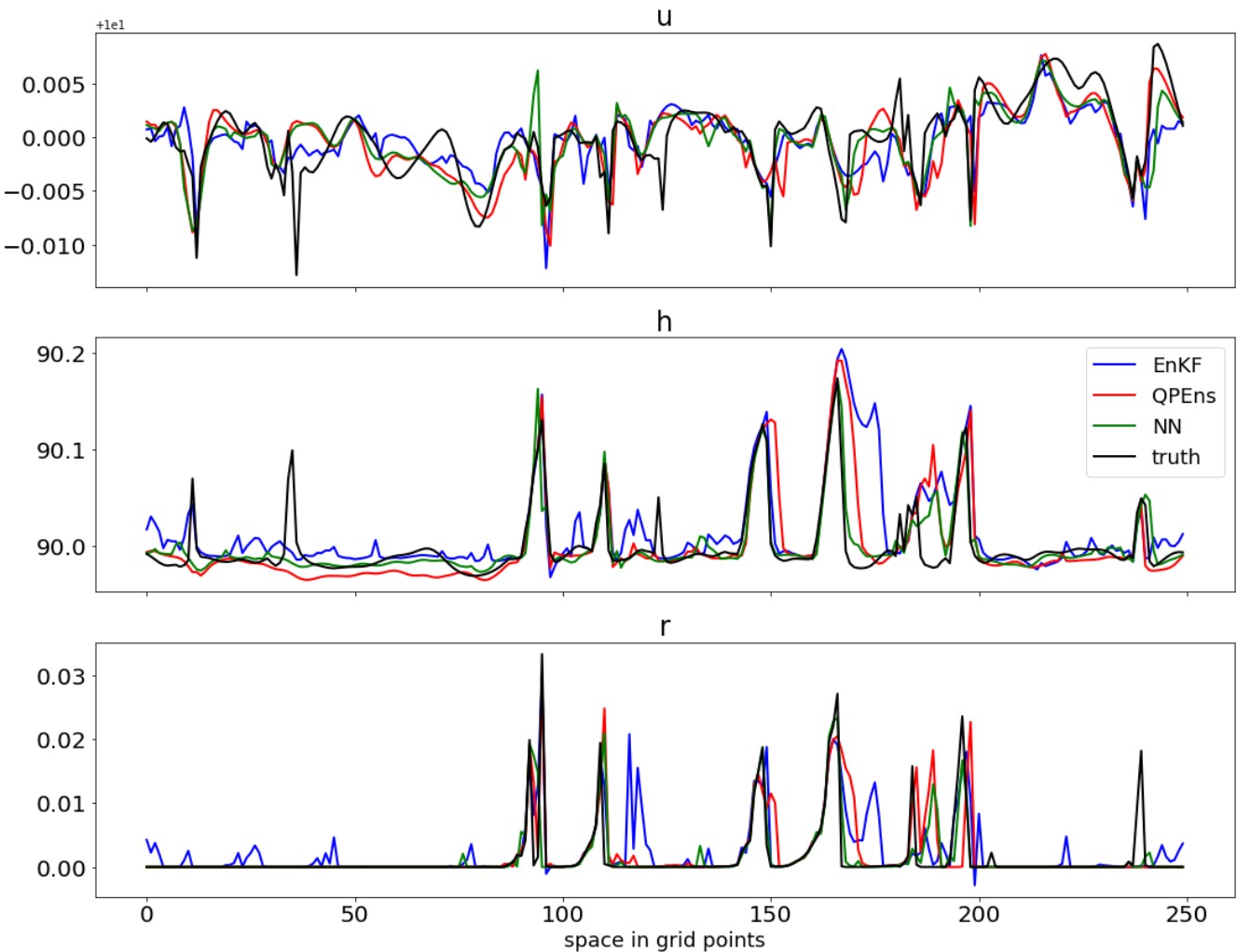

**Figure 8.** Truth (black) and ensemble mean snapshot for EnKF (blue), QPEns (red) and NN with $dT10_{\eta=2}$ (green) before negative rain values are set to zero for the EnKF.

*Acknowledgements.* The research leading to these results has been done within the subproject B6 of the Transregional Collaborative Research Center SFB / TRR 165 "Waves to Weather" (www.wavestoweather.de) funded by the German Research Foundation (DFG). Tijana Janjić is thankful to DFG for funding her research through Heisenberg Programm JA 1077/4-1.

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
