# Peer review of "Training a convolutional neural network to conserve mass in data assimilation"

_Nonlinear Processes in Geophysics, 2020_

## Referee Comment (RC1) · Marc Bocquet (Referee) · 24 Oct 2020

**1   Possible improvements**

This is a nicely written paper with a clear-cut organisation. The paper is convincing and well illustrated. Among possible improvements, I would list:

- The manuscript may be a bit short and could benefit from more in-depth or additional experiments if relevant.

- A few relevant and more recent references could be added (*recent* is very short in this subject).

[Figure]

- It would be much better to make the codes available for the sake of repeatability, as is customary in the machine learning community; maybe not all of them, since that may become tedious, but for instance the model and the machine learning code pieces.

- The line and equations numbering could/should be corrected/improved.

Please see below for the details about these suggestions. Overall, I believe the manuscript only requires minor revisions but that they should be very carefully addressed.

**2  Suggestions and typos:**

1. l.4-6: "In order to produce from a less computationally expensive, unconstrained analysis, a solution that is closer to the constrained analysis, we propose to use a convolutional neural network (CNN) trained on analyses produced by the QPEns.": The sentence is difficult to understand because: (i) there should not be a comma in between "expensive, unconstrained" (ii) "closer": what do you compare to? This is confusing because of the beginning of the sentence; "close" may work better here.

2. l.8-9: "To obtain these positive results, it was in one case necessary to add a penalty term to the loss function of the CNN training process.": This is too vague a statement for an abstract. In my opinion, you should make it more precise or remove it (since the abstract is not long, the former is better).

3. l.17: "Janjić (2016),Zeng et": a space is missing.

4. "Artificial neural networks (NN), are powerful tools" ⟶ "Artificial neural networks (NN) are powerful tools"

5. l.27: "non-linear": nonlinear is much more common (check the title of the journal).

6. l.28: "based on example" $\longrightarrow$ "based on examples"?

7. l.45: Brajard et al. (2019). has actually been accepted as Brajard et al. (2020a). Can you please update the reference?

8. l.36: "combining NN with a knowledge based model as a hybrid forecasting approach (Pathak et al., 2018b)": I believe Brajard et al. (2020b), which recently appeared, is also a very relevant citation to your manuscript because as opposed to Pathak et al. (2018) who rely on only one degree of freedom in model error and reservoir computing, Brajard et al. (2020b) have many degrees of model error freedom and rely on CNNs, like you do.

9. l.75: "Gaussian stochastic forcing $\beta_u$ has a half width of 4 grid points": Is this remark about correlation length of the covariance matrix?

10. l.82: "with parameters $\mu = -8$ and $\sigma = 1.5$.": You have to be more precise. What are $\mu$ and $\sigma$? You know that it can be ambiguous for log-normal distributions (whether you consider the variable of the log-variable).

11. l.87: "using 5-th order polynomial function (Gaspari and Cohn, 1999)": I believe that what you use is actually a 5-th piecewise rational function, is it?

12. l.94-95: "the analysis error is larger than that of an arbitrary model state.": Do you mean larger than the climatological standard deviation of the model state? It's unclear to me.

13. l.117-119: I believe that you should give a reference for the selu activation function because giving those values would seem strange to typical readers of Nonlinear Processes in Geophysics (in particular they cannot really guess that they are meant to be optimal in some sense).

14. l.123-124: "We set the batch size to 96 and do 100 epochs." ⟶ "We set the batch size to 96 and run 100 epochs."?

15. You should have use the latex package linenofix.sty. Your line numbering has issues!

16. Please number all of your equations. This is customary – this facilities the study of your paper by colleagues and students. Systematic numbering may be avoided in reports and book to avoid cluttering.

17. p.5: Equation defining the loss function (no number and line numbers skipped): Why do you take the square root and not the MSE which is available in Tensor-Flow/Keras?

18. l.119: "The python library Keras (Chollet et al., 2015).": (i) You are actually using TensorFlow/Keras or TensorFlow 2.x. – your statement is a bit weird. (ii) Please give the reference to Chollet's book instead, which is the Keras bible as well as an excellent introduction to TensorFlow/Keras and more generally deep learning (Chollet, 2017).

19. It would be better to provide your codes. Maybe not all pieces, but for instance the original ones like the convection model and the TensorFlow code.

20. l.135 and Figure 2: Did you average your RMSEs over several learning and/or test experiments? It is possible that the curves are significantly dependent on the initial random seed. If not, I do not expect any unpleasant surprises but more reliable (and less noisy) curves, potentially with error bars. Please clarify.

21. p.9; Table 2 caption: "As table 1, but for" ⟶ "Same as table 1, but for". Same remark for Figures 5 and 6, and maybe others(?).

22. l.156-165: It may be that the CNN is actually correcting for other sources of model errors such as the impact of localisation. That would explain why EnKF+CNN can outperform QPEns.

23. l.175: the sentences are a bit awkward, I suggest (2 corrections): "the CNN was able to reduce the mass violation significantly. Moreover,"

24. Acknowledgements: There seems to be a useless " at the beginning.

**References**

Brajard, J., Carrassi, A., Bocquet, M., Bertino, L., 2020a. Combining data assimilation and machine learning to emulate a dynamical model from sparse and noisy observations: a case study with the Lorenz 96 model. J. Comput. Sci. 44, 101171. doi:10.1016/j.jocs.2020.101171.

Brajard, J., Carrassi, A., Bocquet, M., Bertino, L., 2020b. Combining data assimilation and machine learning to infer unresolved scale parametrisation. Philosophical Transactions A 0, 0. URL: https://arxiv.org/pdf/2009.04318.pdf. accepted.

Chollet, F., 2017. Deep Learning with Python. Manning Publications Company.

Pathak, J., Wikner, A., Fussell, R., Chandra, S., Hunt, B.R., Girvan, M., Ott, E., 2018. Hybrid forecasting of chaotic processes: using machine learning in conjunction with a knowledge-based model. Chaos 28, 041101. doi:10.1063/1.5028373.

---

## Referee Comment (RC2) · Svetlana Dubinkina (Referee) · 5 Nov 2020

This is a well-written manuscript with very interesting results. My major comment is that this manuscript is rather short and that it could be extended to give more insightful results.

Major comments:

1) I would be in favour to see how the conclusions change depending of the grid size and the ensemble size.

2) There is a trade-off between mass conservation and low RMSE for u and h. What happens if in the experiments with the additional penalty term for mass conservation instead of a linear activation function for u and h, the "relu" activation function is used
for both u and h as well as for r? Is the trade-off smaller then?

3) Authors remove the climatological mean from u and h. What happens if the climatological mean is not subtracted? Is the bias too high for the methods to handle?

Minor comments:

1) I.8: The last sentence of the abstract is rather vague. Please elaborate.

2) I.146: Does the loss function  $\hat{J}_{\gamma}$  account for the mass twice: in J and in the penalty term?

3) Please change  $\gamma$  to something else, since it is already reserved for the gravity wave speed.

4) Why is the penalty term chosen in such a way, namely L1 norm and not L2 as in J?

5) If I look at Fig. 2(a) I see that NN is performing slightly better than QPEns. Is there an explanation for that?

6) I.92: "For the EnKF negative values for rain are set to zero if they occur". This is the variable r, if I understand correctly. However, if I look at Figure 7, I see negative values of r for EnKF. Could authors please explain?

7) A table consistent of wall-clock time for different methods would be insightful for the computational cost gain.

8) I do not want to be self-promoted but authors could have a look at Dubinkina 2018 and decide if they would like to refer to it in their manuscript.

S. Dubinkina, "Relevance of conservative numerical schemes for an Ensemble Kalman Filter", Q.J.R. Meteorol. Soc., 144 (2018), pp.467-477, doi:10.1002/qj.3219

NPGD

---

## Author Comment (AC1) · 15 Dec 2020

article xcolor

**Review response**

December 15, 2020

Thank you for taking the time to thoroughly read our manuscript and for the positive feedback. We agree with your comments and have adjusted the manuscript accordingly, see below.

**1 Possible improvements**

This is a nicely written paper with a clear-cut organisation. The paper is convincing and well illustrated. Among possible improvements, I would list:

- The manuscript may be a bit short and could benefit from more in-depth or additional experiments if relevant.
  We performed additional experiments to investigate the trade off between mass conservation and RMSE, which are now summarized in Figure 4. Note that we have changed the definition of the penalty term by comparing the mean fields of h, not the sum, so the penalty term is divided by n=250 now.

- A few relevant and more recent references could be added (recent is very short
in this subject).
We added the following references:

- – Bocquet, M., Brajard, J., Carrassi, A., and Bertino, L.: Bayesian inference of chaotic dynamics by merging data assimilation, machinelearning and expectation-maximization, Foundations of Data Science, 2, 55–80, https://doi.org/10.3934/fods.2020004, 2020.
- – Brajard, J., Carrassi, A., Bocquet, M., and Bertino, L.: Combining data assimilation and machine learning to infer unresolved scale parametri-sation, URL:https://arxiv.org/pdf/2009.04318.pdf, 2020b
- – Farchi, A., Laloyaux, P., Bonavita, M., and Bocquet, M.: Using machine learning to correct model error in data assimilation and forecastapplications, 2020
- – Watson, P. A. G.: Applying Machine Learning to Improve Simulations of a Chaotic Dynamical System Using Empirical Error Correction,Journal of Advances in Modeling Earth Systems, 11, 1402–1417, https://doi.org/https://doi.org/10.1029/2018MS001597, 2019
- – Yuval, Janni and O'Gorman, Paul A: Stable machine-learning parameterization of subgrid processes for climate modeling at a range of resolutions,Nature communications, 11, 1, 1–10, 2020, Nature Publishing Group
- – Stephan Rasp and Nils Thuerey:Data-driven medium-range weather prediction with a Resnet pretrained on climate simulations: A new model for WeatherBench, 2020,arXiv preprint arXiv:2008.08626

- It would be much better to make the codes available for the sake of repeatability, as is customary in the machine learning community; maybe not all of them, since that may become tedious, but for instance the model and the machine learning code pieces.
  We will provide the code.
- The line and equations numbering could/should be corrected/improved.
  fixed

Please see below for the details about these suggestions. Overall, I believe the manuscript only requires minor revisions but that they should be very carefully addressed.

**2 Suggestions and typos:**

1. l.4-6: "In order to produce from a less computationally expensive, unconstrained analysis, a solution that is closer to the constrained analysis, we propose to use a convolutional neural network (CNN) trained on analyses produced by the QPEns.": The sentence is difficult to understand because: (i) there should not be a comma in between "expensive, unconstrained" (ii) "closer": what do you compare to? This is confusing because of the beginning of the sentence; "close" may work better here.
   We rephrased to "We therefore propose to use a convolutional neural network (CNN) trained on the difference between the analysis produced by a standard ensemble Kalman Filter (EnKF) and the QPEns to correct any violations of imposed constraints."

2. l.8-9: "To obtain these positive results, it was in one case necessary to add a penalty term to the loss function of the CNN training process.": This is too vague a statement for an abstract. In my opinion, you should make it more precise or remove it (since the abstract is not long, the former is better).
   We removed it.

3. l.17: "Janjic (2016),Zeng et": a space is missing. '
   fixed

4. "Artificial neural networks (NN), are powerful tools" $-\rightarrow$ "Artificial neural networks (NN) are powerful tools"
   fixed

5. l.27: "non-linear": nonlinear is much more common (check the title of the journal).
   fixed

6. l.28: "based on example" $-\rightarrow$ "based on examples"?
   fixed

7. l.45: Brajard et al. (2019). has actually been accepted as Brajard et al. (2020a). Can you please update the reference?
   fixed

8. l.36: "combining NN with a knowledge based model as a hybrid forecasting approach (Pathak et al., 2018b)": I believe Brajard et al. (2020b), which recently appeared, is also a very relevant citation to your manuscript because as opposed to Pathak et al. (2018) who rely on only one degree of freedom in model error and reservoir computing, Brajard et al. (2020b) have many degrees of model error freedom and rely on CNNs, like you do.
   We added the reference

9. l.75: "Gaussian stochastic forcing $\beta_u$ has a half width of 4 grid points": Is this remark about correlation length of the covariance matrix?
   No, a Gaussian shaped term $\beta_u$ is added to the wind field at each model time step at a random location (see line 72 of the new manuscript).

10. l.82: "with parameters $\mu = -8$ and $\sigma = 1.5$.": You have to be more precise. What are $\mu$ and $\sigma$? You know that it can be ambiguous for log-normal distributions (whether you consider the variable of the log-variable).
    Good point. We rephrased: "a lognormal error is added to the rain field with parameters of the underlying normal distribution $\mu=-8$ and $\sigma= 1.5$"

11. l.87: "using 5-th order polynomial function (Gaspari and Cohn, 1999)": I believe that what you use is actually a 5-th piecewise rational function, is it? Thank you. We fixed it.

12. l.94-95: "the analysis error is larger than that of an arbitrary model state.": Do you mean larger than the climatological standard deviation of the model state? It's unclear to me.
Yes, we have rephrased.

13. l.117-119: I believe that you should give a reference for the selu activation function because giving those values would seem strange to typical readers of Nonlinear Processes in Geophysics (in particular they cannot really guess that they are meant to be optimal in some sense).
We have added a reference: "These values are chosen such that the mean and variance of the inputs are preserved between two consecutive layers (Klambauer et al., 2017)"

14. l.123-124: "We set the batch size to 96 and do 100 epochs." −→ "We set the batch size to 96 and run 100 epochs."?
fixed

15. You should have use the latex package linenofix.sty. Your line numbering has issues!
We use the Copernicus Publications Manuscript Preparation Template for LaTeX Submissions. Now that all equations are numbered, the line numbering is also fixed.

16. Please number all of your equations. This is customary – this facilities the study of your paper by colleagues and students. Systematic numbering may be avoided in reports and book to avoid cluttering.
Agreed, we have fixed this.

[Figure]

17. p.5: Equation defining the loss function (no number and line numbers skipped): Why do you take the square root and not the MSE which is available in Tensor-Flow/Keras?
Because we also look at RMSE when verifying the data assimilation results, so this was just an easier direct comparison. We checked that using the MSE of TensorFlow/Keras yields similar results. However, the tables look very different when expressed in terms of MSEs instead of RMSEs. For example, the improvement in terms of RMSE is 32%, whereas in terms of MSE it is 59%.

18. l.119: "The python library Keras (Chollet et al., 2015).": (i) You are actually using TensorFlow/Keras or TensorFlow 2.x. – your statement is a bit weird. (ii) Please give the reference to Chollet's book instead, which is the Keras bible as well as an excellent introduction to TensorFlow/Keras and more generally deep learning (Chollet, 2017).
fixed

19. It would be better to provide your codes. Maybe not all pieces, but for instance the original ones like the convection model and the TensorFlow code.
We will provide the code.

20. l.135 and Figure 2: Did you average your RMSEs over several learning and/or test experiments? It is possible that the curves are significantly dependent on the initial random seed. If not, I do not expect any unpleasant surprises but more reliable (and less noisy) curves, potentially with error bars. Please clarify.
We averaged over 500 experiments. This information is now included in the Figure captions.

21. p.9; Table 2 caption: "As table 1, but for" $\longrightarrow$ "Same as table 1, but for". Same remark for Figures 5 and 6, and maybe others(?).
fixed

22. l.156-165: It may be that the CNN is actually correcting for other sources of model errors such as the impact of localisation. That would explain why EnKF+CNN can outperform QPEns.
Yes, that is a good point since the CNN has an influence radius of 5 grid points and the data assimilation a radius of 8 grid points. We therefore trained an additional CNN with the kernel of all layers of size 5, so that the influence radius is 10. This gave us however similar results as in Figure 5 and 6. We added this discussion in text: "Since the CNN only has an influence radius of 5 grid points and the localisation cut-off radius of the data assimilation is 8 grid points, it is possible that the better results of the CNN stem from this shorter influence radius. However, a CNN trained on the same data but with kernel sizes of 5 instead of 3 (leading to an influence radius of 10 grid points) yields similar results as in Figures 5 and 6 (not shown)."

23. l.175: the sentences are a bit awkward, I suggest (2 corrections): "the CNN was able to reduce the mass violation significantly. Moreover,"
fixed

24. Acknowledgements: There seems to be a useless " at the beginning.
fixed

---

## Author Comment (AC2) · 15 Dec 2020

article xcolor

[Figure]

**Review response**

December 15, 2020

This is a well-written manuscript with very interesting results. My major comment is that this manuscript is rather short and that it could be extended to give more insightful results.

Thank you for taking the time to thoroughly read our manuscript and for the positive feedback. We have taken your comments into account and have adjusted the manuscript accordingly, see below.

Major comments:

1. I would be in favour to see how the conclusions change depending of the grid size and the ensemble size.

   The relative improvement of the QPEns over EnKF for different DA settings (including ensemble size) has been covered in Ruckstuhl and Janjic (2018). We added a sentence at the end of section 2.2: "We refer to Ruckstuhl and Janjic (2018) for a comparison of the performance of the EnKF and the QPEns as a function of ensemble size for different localisation radii, assimilation windows and observation coverage."

We are confident that as long as the CNN can remove the bias in h, the CNN can match the performance of QPEns for any DA setting. One could then try to compare the differences in training process of the CNN's (does one setting require more data than the other?). However, a clean comparison among the different settings would require rigorous tuning of the architecture, the amount of training data needed, and the training process of the CNN. And this tuning is very tricky because we are interested in the performance of the CNN in DA context, not the value of the loss function. Since we are working in a highly idealised setup, we want to be economical with time spent on fine tuning the CNN's. We therefore feel that we have exploited the modified shallow water model on this specific topic. Any further experiments should be done on more complex models. That being said, we did investigate in addition the trade off between mass and RMSE as you suggested in the next point and performed additional experiments that test the relation of the kernel size of the CNN to localisation.

2. There is a trade-off between mass conservation and low RMSE for u and h. What happens if in the experiments with the additional penalty term for mass conservation instead of a linear activation function for u and h, the "relu" activation function is used for both u and h as well as for r? Is the trade-off smaller then? The reason we use the relu function for r is that the rain cannot be negative. This does not hold for u and h, so using the relu function for these variables is not an option. We did perform some additional experiments to investigate the trade off between mass conservation and RMSE, which is now summarized in Figure 4.

3. Authors remove the climatological mean from u and h. What happens if the climatological mean is not subtracted? Is the bias too high for the methods to handle? We want to clarify that we only subtract the mean to make the problem better conditioned for the training process. Since the difference between input and output data for the 3 variables differ in at most 1 order of magnitude, we do not expect huge problems if the mean is not subtracted. However, as far as we know, there

is no disadvantage to normalizing the training data, which is why we have not tried training the CNN on the raw data.

Minor comments:

1. l.8: The last sentence of the abstract is rather vague. Please elaborate.
   We have removed this sentence.

2. l.146: Does the loss function JĔĘ $\gamma$ account for the mass twice: in J and in the penalty term?
   J is the RMSE averaged over the 3 variables. This means that J accounts for the mass error indirectly (as the RMSE goes to zero, the mass error also goes to zero). The penalty term directly accounts for mass errors by first averaging the h field over the 250 grid points for $y^p$ and $y$ separately, and then squaring the difference.

3. Please change $\gamma$ to something else, since it is already reserved for the gravity wave speed.
   Yes, you are right. We changed it to $\eta$.

4. Why is the penalty term chosen in such a way, namely L1 norm and not L2 as in J?
   Note that J takes the norm of a vector of size 250, whereas the penalty term takes the norm of a scalar (namely the difference of the spatial mean of h). Therefore the L1 and L2 norm are equivalent for the penalty term.

5. If I look at Fig. 2(a) I see that NN is performing slightly better than QPEns. Is there an explanation for that?
   The QPEns is also not perfect, so it is possible that the CNN performs better. It is indeed then interesting to speculate why that is. Most of the last paragraph before the conclusion is dedicated to this (from line 174 to 184 in the new manuscript).

6. l.92: "For the EnKF negative values for rain are set to zero if they occur". This is the variable r, if I understand correctly. However, if I look at Figure 7, I see negative values of r for EnKF. Could authors please explain?
   The fields are shown before negative values are set to zero. We have clarified this in the caption

7. A table consistent of wall-clock time for different methods would be insightful for the computational cost gain.
   The costs of applying the NN are negligible with respect to the costs of the EnKF, as mentioned in line 50 of the new manuscript. So it is about the difference in computational costs between the EnKF and QPEns. Since we are working with a cheap model, no effort has been made in the implementation of the algorithms to make them computationally efficient. Therefore wall-clock times may be misleading. However we agree this is an important point and we actually have a paper under review that thoroughly discusses the computational costs of the QPEns. We therefore added a sentence in the introduction: "For a detailed discussion on the computational costs of the QPEns we refer to Janjic et al. (under review)".

8. I do not want to be self-promoted but authors could have a look at Dubinkina 2018 and decide if they would like to refer to it in their manuscript. Thanks for mentioning this paper. We added now a reference to this manuscript.

---

## Author Response (AR2)

thank you for your revised version and for the exhausting answers to the Reviewers. I found those, and the changes applied to the original version of the manuscript, very pertinent.
I consider the current almost ready to be accepted, but I would like the Authors to slightly improve Sect.2.3 which I still found a bit unclear.
If you wish, please use a simple illustration to describe the different datasets. However please do consider expanding a bit on the reasons behind the choice of "I" and of the radar observations positions.

*Thank you for the positive feedback. We have included a schematic of the generation of the data sets (see below). We also included an explanation of why we use I_t:*

*"We include this information because we know from Ruckstuhl and Janjic (2018) that the strength of the QPEns lies in suppressing spurious convection. Since the radar observations cover only rainy regions, the data set I_t can help the CNN to distinguish between dry and rainy regions and possibly develop a different regime for each situation. We verified that the CNN yields significantly different output when setting all values of I_t to zero, indicating that the CNN indeed uses this information. We are open to more suggestions on how to clarify!*